# Techniques for Dealcoholization of Wines: Their Impact on Wine Phenolic Composition, Volatile Composition, and Sensory Characteristics

**DOI:** 10.3390/foods10102498

**Published:** 2021-10-18

**Authors:** Faisal Eudes Sam, Teng-Zhen Ma, Rafia Salifu, Jing Wang, Yu-Mei Jiang, Bo Zhang, Shun-Yu Han

**Affiliations:** Gansu Key Laboratory of Viticulture and Enology, College of Food Science and Engineering, Gansu Agricultural University, Lanzhou 730070, China; sameudes0@gmail.com (F.E.S.); matz@gsau.edu.cn (T.-Z.M.); salifurafiat@gmail.com (R.S.); wj2296@163.com (J.W.); jym316@126.com (Y.-M.J.); zhangbo@gsau.edu.cn (B.Z.)

**Keywords:** dealcoholization, reduced-alcohol wine, alcohol-free wine, non-alcoholic wine, phenolic composition, volatile composition, aroma compounds, sensory quality

## Abstract

The attention of some winemakers and researchers over the past years has been drawn towards the partial or total dealcoholization of wines and alcoholic beverages due to trends in wine styles, and the effect of climate change on wine alcohol content. To achieve this, different techniques have been used at the various stages of winemaking, among which the physical dealcoholization techniques, particularly membrane separation (nanofiltration, reverse osmosis, evaporative perstraction, and pervaporation) and thermal distillation (vacuum distillation and spinning cone column), have shown promising results and hence are being used for commercial production. However, the removal of alcohol by these techniques can cause changes in color and losses of desirable volatile aroma compounds, which can subsequently affect the sensory quality and acceptability of the wine by consumers. Aside from the removal of ethanol, other factors such as the ethanol concentration, the kind of alcohol removal technique, the retention properties of the wine non-volatile matrix, and the chemical-physical properties of the aroma compounds can influence changes in the wine sensory quality during dealcoholization. This review highlights and summarizes some of the techniques for wine dealcoholization and their impact on wine quality to help winemakers in choosing the best technique to limit adverse effects in dealcoholized wines and to help meet the needs and acceptance among different targeted consumers such as younger people, pregnant women, drivers, and teetotalers.

## 1. Introduction

Wine is an alcoholic beverage popularly produced from fermented grape juice. Wines can be classified as red, rose (pink), or white based on their color, and they can also be classified as table (red, rose, or white), sparkling, or fortified based on their alcohol level or carbondioxide content [1]. Table wines are wines that are neither fortified nor sparkling and are typically served with food [2]. Fortified wines are made by adding alcohol (usually between 16% and 23%) [1,2,3]. Wines can also be classified based on how much carbon dioxide they contain. Those that contain carbon dioxide (about 10 g/L CO_2_) [4] are classified as sparkling wines, while those that do not contain carbon dioxide are classified as “still” wines [1]. The carbon dioxide can be produced naturally during fermentation or added artificially. Based on alcoholic content, wines can further be classified as alcohol-free (< 0.5% v/v), low-alcohol (0.5% to 1.2% v/v), reduced-alcohol (1.2% to 5.5% or 6.5% v/v), lower-alcohol (5.5% to 10.5% v/v), and alcoholic wines (> 10.5% v/v) [5,6]. In addition, wines are also classified according to their sugar content: dry (maximun of 4 g/L sugar), medium dry (between 4 g/L and 12 g/L sugar), semi-sweet (between 12 g/L and 45 g/L sugar), and sweet (minimum of 45 g/L sugar) [7]. However, these classifications are not explicit and may vary between most wine producing countries and the applicable legislations. In the UK, for example, wines with an alcohol content of 1.2% alcohol by volume (ABV) or less are classified as low alcohol wines, while wines with an alcohol content of less than 0.5% ABV are referred to as non-alcoholic wines. In contrast, China classifies low alcohol wines as wines with 1.0% to 7.0% ABV and non-alcoholic wines as wines with 0.5% to 1.0% ABV [8].

From several studies (in vitro and in vivo), there is a positive consent of the beneficial impact of wine consumption on neurological diseases, cardiovascular disease, osteoporosis, diabetes, and longevity [9,10,11,12,13,14]. When consumed in adequate amounts and together with a meal, wine plays a vital role in mitigating oxidative stress and vascular endothelial damage induced by a high-fat meal [15]. According to Boban et al. [15], red wine consumption may help prevent heart diseases as well as type two diabetes, allowing consumers to enjoy better health and an increased lifespan as they age. A Chinese study on alcohol and mortality in middle-aged men discovered a 19% reduction in deaths with no more than two drinks per day [16]. Furthermore, a study conducted by Buettner and Skemp [17] on blue zones revealed adequate wine intake as one of the nine lifestyle habits in populations around the world that are known for their long lifespan and healthy aging. Despite the benefits associated with wine consumption, some consumers perceive wine to be harmful to human health because it contains alcohol [18].

High concentrations of ethanol in wine increase the sensation of hotness and bitterness, while decreasing acidity and masking the sensitivity of certain essential aroma compounds such as esters, higher alcohols, and monoterpenes [19,20,21]. Furthermore, high alcohol wines are subject to higher import duties and taxes in some countries [22]. For example, in the United States, wine with 14% alcohol or less is taxed at USD 1.07 per gallon, while wine with 14.1% to 21% alcohol is taxed at USD 1.57 per gallon [23]. There is a common view all over the world that the consumption of alcoholic wine should lessen in favor of low or non-alcoholic wines [24,25,26]. This is currently being witnessed globally as there is a growing popularity of low- or non-alcoholic wines and beverages, particularly in Europe and North America (www.factmr.com/report/4532/non-alcoholic-wine-market, accessed on 1 September 2021). Consumer preferences are shifting with consumers in the non-alcoholic wine market wanting new product offerings and alternatives. There is also an increasing percentage of the adult population seeking lower alcohol wines and beverages more frequently, which has boosted non-alcoholic wine sales. This trend has prompted producers to introduce new non-alcoholic wine products with fruity and floral notes. Additionally, the global non-alcoholic wine market size is valued at USD 20 billion with a compound annual growth rate (CAGR) of over 45% in 2018 and is projected to increase at a remarkable CAGR of over 7% during the forecast period (2019–2027), reaching a value pool of over USD 30 billion [24]. According to another school of thought (www.factmr.com/report/4532/non-alcoholic-wine-market, accessed on 1 September 2021), the global market will continue to grow steadily, with a CAGR of 10.4% from 2021 to 2031, up from an 8.8% CAGR from 2016 to 2020. Therefore, for wine producers to meet consumers’ demands and adapt to the risisng non-alcoholic wine market, they need to produce high-quality alcohol-free or low-alcoholic wines.

Fortunately, over the last 15 to 20 years, winemakers and researchers have developed various techniques (Figure 1) for reducing the alcohol content of wines and beverages to meet consumers’ demand for low-alcohol, reduced-alcohol, or non-alcoholic wines and beverages with good sensory qualities, as well as to adapt to the risisng non-alcoholic wine market. These techniques can be classified according to the stage of wine production, thus, the pre-fermentation stage, fermentation stage, and post-fermentation stage [5,27,28,29,30]. The reduction of alcohol at the pre-fermentation, fermentation, and post-fermentation stages include the reduction of fermentable sugars, the reduction of alcohol production, and separation by a membrane or non-membrane apparatuses, respectively. However, the techniques used during the pre-fermentation and fermentation stages usually result in wines with alcohol contents that are still above the required ethanol concentration (0.5% to 1.2% v/v) needed to fall under low-alcohol or alcohol-free wines [5]. In situations whereby this alcohol reduction in wines results in low-alcohol or alcohol-free wines, such wines usually have poor sensory qualities [31,32]. However, some well-established post-fermentation techniques, such as reverse osmosis (RO) and osmotic distillation (OD) have been used for the removal of alcohol up to content lower than 5% v/v without significantly changing the main quality parameters of the wine [33,34,35,36]. This review article comprehensively summarizes the techniques used for the dealcoholization of various wines, particularly the physical dealcoholization techniques that can result in a high reduction of wine alcohol content. In addition, we discuss the impact of alcohol reduction by physical techniques on wine quality, specifically the phenolic composition, volatile composition, and sensory characteristics.

## 2. Techniques for Wine Alcohol Reduction

A summary of some techniques commonly used for the dealcoholization of wines at the various stages (pre-fermentation stage, fermentation stage, and post-fermentation stage) of wine production and their extent of ethanol removal is shown in Table 1.

### 2.1. Limiting Alcohol Production during Pre-Fermentation Stage

#### 2.1.1. Viticultural Techniques

Climate change has resulted in the production of grapes with higher sugar levels over the last 20 years [37]. The accumulation of sugar, primarily glucose and fructose, within the cellular medium, particularly in the vacuoles, is a key feature of the ripening process in grape berries and has a significant impact on the alcohol content in wines. Glucose accounts for half of the total fermentable sugars in grape juice [28], and it is the primary substrate in grape must that yeast converts into alcohol during fermentation. Alcohol production can be controlled during the pre-fermentation stage by lowering the concentration of fermentable sugars in juice through viticultural practices such as leaf area reduction, modified irrigation systems, applying growth regulators, and photosynthetic activity reduction [38,48,49,50,51].

One of the most important viticultural indices for defining a well-balanced vineyard capable of producing high-quality grapes and wine is leaf area reduction. It is possible to achieve this by performing severe trimming or leaf removal at various stages of berry growth. Leaf area reduction has shown to have a strong influence on the rate of sugar accumulation in berries [38,49,52,53], which can lower the alcohol concentration in the resulting wine [51,54,55].

Grape sugar concentration rises with an increasing leaf area to fruit ratio and eventually reaches a plateau [39,40,41,42]. According to Kliewer and Dokoozlian [42], grapes can achieve good ripeness with a leaf area to fruit ratio ranging from 0.8 to 1.2 m^2^/kg. If the leaf area to fruit ratio is high, the sugar concentration may rise to an unacceptable level by the time the flavor and phenolic ripeness for a specific wine style occurs [43]. The reduction in leaf area caused by the apical defoliation of Shiraz grapevines resulted in a lower alcohol level in the finished wine [44]. A mechanical leaf removal method was used on ‘Sangiovese’ vines four weeks after veraison above the cluster zone to delay sugar accumulation, which was effective in slowing down berry sugaring and achieving less alcoholic wines (reduction of 0.6% v/v) [53]. Poni et al. [41] obtained comparable results when they evaluated a late leaf removal above a Sangiovese bunch area (at post veraison, average 12 °Brix). In this manner, the total soluble solids content in grape must and wine alcohol concentration were significantly reduced (the latter by 0.6% v/v) without affecting other compositional parameters such as phenolic substances.

Stoll et al. [50] adjusted the vine leaf area to crop ratio in the early stage of vine growth to reduce the concentration of fermentable sugars in grapes during maturation, which proved to be a promising approach in managing grape sugar concentrations. Filippetti et al. [45], by late-topping ‘Sangiovese’ shoots one week after veraison, also obtained a good reduction in must sugar concentration without modifying the pH, organic acid content, anthocyanins concentration, and skin and seed tannins content. Furthermore, Martínez De Toda et al. [38] discovered a 20-day delay in grape ripening, a lower pH (0.1 to 0.3), a 14% reduction in soluble solids (from 24.4 to 21.0 °Brix), and a 10% to 27% reduction in total anthocyanin content through shoot trimming of Grenache over 3 years, which resulted in a 2% v/v reduction in wine alcohol. Sun et al. [51], on the other hand, observed an increase in the soluble solids of 0.8 °Brix and a subsequent 0.7% v/v increase in alcohol content after applying shoot thinning to Corot Noir wine grapes. Regardless of the cultivar or vine productivity, the use of growth regulators can result in a significant reduction in must sugar concentration and wine alcohol content [39,46,47,50]. Böttcher et al. [46] used an anti-transpirant (1-naphthalene acetic acid) on *Vitis vinifera* L. cv. Syrah grapes during pre-veraison, which effectively delayed the start of berry ripening and improved sugar accumulation management without affecting wine sensory characteristics. However, anti-transpirant treatments may have a negative impact on phenolic content (particularly in red grape varieties) and anthocyanins, while the total polyphenol content appears to be unaffected [47,50].

#### 2.1.2. Early Grape Harvest

Another pre-fermentation strategy that can be used to reduce wine alcohol content is harvest date management. Some red and white grape varieties have been studied in low-alcohol wine production through the early harvesting of grapes or the blending of matured grapes with early harvested grapes [56,57,59,60,61,64,65]. Unripe red or white grapes can be used to make a low-alcohol, high-acidity blending material, which is then added to a more mature fruit ferment. This method was reported to result in a 3% v/v decrease in ethanol concentration, however, the wines exhibited undesirable acidic and unripe flavors [58,59]. Piccardo et al. [64] substituted grape juice obtained from two red grape varieties (Pinot Noir and Tannat) with varying degrees of maturity to produce red wines with lower alcohol content and pH and higher color and phenolic compound concentrations. Wines made from the substituted juice had lower alcohol content, pH, and total acidity, but no significant differences in other wine components [64]. The alcohol content and aromatic expression of Barbera and Pinot Noir wines made with early harvested grapes were also evaluated, where the earlier harvesting of grapes resulted in a reduction in wine alcohol (−3.2% v/v), especially for Barbera wine with optimal aromatic profiles [60].

#### 2.1.3. Dilution/Blending of Grape Must

Wine alcohol content can be reduced simply by adding water to grape must to dilute the fermentable sugar concentrations. Though, in some countries such as the United States (excluding California), water can be added in very small amounts as a processing aid to achieve the desired result, this act is illegal and restricted in most other countries like South Africa, New Zealand, Austrilia, France, and Germany [27,72]. Some studies have reported good results from adding water to grape must to reduce wine alcohol content [61,62,63,67,68,70,71]. For example, Schelezki et al. [62] investigated the pre-fermentative substitution of Cabernet Sauvignon grape must (30.4 °Brix) with either “green harvest wine” of 4.5% ABV or water in the production of lower alcohol wines. When compared to the control wine (18.1% ABV), this approach resulted in wines with lower alcohol content (14.5% ABV), with no significant effect on wine chemistry (including total anthocyanins, wine color density, and total phenolics). The sulfur-dioxide-resistant pigments, on the other hand, decreased only at the highest substitution levels for water or “green harvest wine” additions. Harbertson et al. [73] reported a 2% (v/v) reduction in the alcohol content of a wine produced from diluted Merlot must. The wine was found to have an increased fresh fruit flavor with no change in perceived heat sensation compared to the control wine. Similarly, wine produced by diluting Cabernet Sauvignon must with water from 30 to 24 °Brix showed a reduction in alcohol content with similar sensory results as wine obtained from early grape harvest [68]. However, higher levels of water addition may reduce must acidity, lower the concentrations of phenolics and tannins, and have a negative impact on wine appearance and sensory attributes [63,69,70,71].

#### 2.1.4. Filtration of Grape Juice

The filtration of grape must prior to fermentation for sugar reduction is another method for lowering wine alcohol content. Several studies have investigated the use of membrane technology for sugar removal from must before fermentation [74,75,76,77,79,80,81]. García-Martín et al. [74] investigated sugar reduction in Verdejo and Tinta de Toro grape musts using a single and two-step nanofiltration (NF) process. The untreated must was mixed with the filtration permeate and retentate obtained from the nanofiltration process in various proportions sufficient to reduce the alcohol content of the resulting wines by 2% v/v. The alcohol content of the resulting wines was reduced by 3.3% v/v after fermentation. However, there was a slight decrease in color intensity and some volatile compounds, as well as less perception of floral and fruity aromas. Similarly, Salgado et al. [79] investigated a single-stage and two-stage NF process with a 200 Da spiral-wound membrane to decrease the sugar content of white and red musts prior to fermentation. The control musts, as well as the filtration permeate and retentate were blended and fermented. When compared to the control wines, wines obtained after the fermentation of the blend must show about 1 to 2 degrees of alcohol reduction with no significant differences in sensory attributes. Furthermore, Mira et al. [77] before fermentation, used a reverse osmosis (RO) process for the filtration of Tinta Roriz, Syrah, and Alicante Bouschet grape must (23.7 °Brix, about 15.2% v/v alcohol) to control the alcohol level in the resulting wine. The resulting wine had up to a 5% v/v reduction in alcohol, but with lower color intensity, polyphenols, and anthocyanins, which influenced the sensory properties. Musts are complex liquids with extreme colloidal properties, which can cause membrane fouling or flux decay during must filtration [74,79,82,83]. Therefore, optimal operating conditions and a suitable membrane conformation with good molecular weight cut-off (MWCO) could increase the retention of volatile compounds and preserve the good taste in wine.

#### 2.1.5. Addition of Enzyme (Glucose Oxidase)

Glucose oxidase, an enzyme derived from the fungus *Aspergillus niger*, can be used to remove grape must glucose before fermentation [84,87]. In a first step reaction, glucose oxidase can convert β-D-glucose to D-glucono-lactone, producing hydrogen peroxide, and in a second step reaction, it can catalyze the conversion of D-glucono-lactone to gluconic acid, producing gluconic acid. These reactions result in the oxidation of must sugar, which prevents the formation of alcohol from the must sugar [89]. Several studies have reported the use of glucose oxidase in wine alcohol reduction [84,85,86,87,88,89]. Pickering [89] used 2 g/L of glucose oxidase to produce reduced-alcohol white wine from Riesling grape juice. The results showed that after 6 hours of fermentation, about 87% of the glucose was converted into gluconic acid, indicating an alcohol reduction of up to 4.3% v/v. When Gluzyme, an *Aspergillus oryzae* derived glucose oxidase was used at a concentration of 30 kU in the treatment of Pinotage grape must before fermentation. It resulted in a 0.7% v/v ethanol decrease when compared to control wines [84]. Furthermore, the acetic acid concentration was found to be lower in the treated wines when compared to the control wines [84]. Additionally, Röcker et al. [87] treated a white grape must with glucose oxidase enzyme and achieved a 2% v/v alcohol reduction, but the treated wines were significantly more acidic and less fruity.

### 2.2. Reducing Alcohol Production during Fermentation Stage

#### 2.2.1. Use of Non-*Saccharomyces cerevisiae* Yeasts

The use of non-*Saccharomyces* (NS) yeasts in wine alcohol reduction have recently attracted the interest of researchers and winemakers. Some NS yeast strains can ferment less sugar or divert carbon metabolism to other pathways, preventing excessive ethanol production during fermentation [95]. The use of NS yeasts in single-strain fermentation, mixed fermentation, or sequential fermentation with *Saccharomyces cerevisiae* have been investigated for an ethanol reduction ranging from 0.2% to 2% v/v [90,91,92,93,94,95,96,97,98,99,100,101,102,103,104,105,106,107,108,109,110,111,112,113,114,115,116,117,118,119,120]. In addition to the potential contribution of NS yeasts to moderate ethanol reduction in wine, some studies have shown that NS yeasts have the potential to improve wine fermentation, sensorial complexity, and the aroma profile [98,108,109,110,111,112]. Furthermore, studies have shown that some NS yeast species or strains can combat spoilage yeasts [113,114,115], which is an added advantage.

#### 2.2.2. Use of Modified Yeast Strains

Metabolic engineering through gene modification techniques or adaptive evolution and selection has led to the development of modified yeast strains that can reduce the ethanol content of wine during fermentation [121,122,123,124,125,126,127,128,129,130,131,132]. Heux et al. [126] genetically expressed an *enoxE* gene encoding an NADH oxidase into a *Saccharomyces cerevisiae* yeast strain (V5) derived from Champagne wine yeast, resulting in significant ethanol, glycerol, succinate, and hydroxyglutarate production. In addition, Hou et al. [132], by introducing five different genes (*nox*, *AOX1*, cytosolic *POS5*, mitochondrial *POS5*, and *udhA*) into a *Saccharomyces cerevisiae* strain CEN.PK113-5D (*MATa SUC2 MAL8C ura3-52*), reduced the mitochondrial and cytosolic NADH levels, resulting in lower glycerol and ethanol production. Despite their modest ability to lower alcohol content, genetically modified *Saccharomyces cerevisiae* yeast strains can also produce acetate, acetaldehyde, and acetoin [126,131], which can negatively affect wine sensory quality. Another disadvantage of using modified yeast strains to lower the alcohol content of wine is consumer rejection of the use of genetically modified organisms (GMOs) in food and beverage production. Therefore, non-genetic techniques such as adaptive evolution and selection, as well as the use of lower ethanol yielding yeasts [123,129] are used, which can reduce wine alcohol content by up to 3% v/v without compromising wine quality [30,124].

#### 2.2.3. Biomass Reduction

Biomass reduction involves reducing the yeast population during must fermentation so that the fermentation rate is reduced. The low fermentation of fermentable sugars can be influenced by biomass reduction, resulting in products with low ethanol content. Some authors [133,134,135] investigated the production of low-alcohol cider by biomass reduction using centrifugation. The results of these studies show that this technique can produce cider with low alcohol content (reduction of up to 4 degrees) and fruity flavor. However, this technique usually results in fermented beverages with significant residual sugars that are microbiologically unstable and can lead to spoilage [136]. In contrast, biomass removal can be very efficient in producing a sweeter and more enjoyable alcoholic beverage [134].

#### 2.2.4. Arrested or Limited Fermentation

Another method of producing low or non-alcoholic wines and beverages is by arrested or limited fermentation. In this method, alcoholic fermentation is deliberately stopped before it is complete by controlling the fermentation temperature and time [89,171,172,173,174,175]. The short fermentation time used in this method is a major drawback, as it is insufficient to adequately convert most of the fermentable must sugars, resulting in sweet wines with high residual sugar content [136]. Wines with difficult microbiological stabilization and storage may require additional finishing procedures such as sulfur dioxide (SO_2_) preservation or thermal pasteurization to extend shelf life [89,136]. In addition, the short fermentation time limits the production of most desirable odor-active volatiles such as monoterpenes, ethyl esters, and acetates, which yeasts produce in large quantities during long fermentation times [176,177,178,179], resulting in non-alcoholic or low-alcohol sweet wines with poor aroma profiles.

### 2.3. Reducing Alcohol Production during Post-Fermentation 

#### 2.3.1. Nanofiltration (NF)

NF is a pressure-driven separation process that uses a semi-permeable membrane with pores ranging from 1 to 10 nm in size [180]. The loose pores of NF membranes allow higher permeate fluxes than RO membranes and provide better rejection for smaller molecules such as sugars, peptides, proteins, etc., at about 75 bar than ultrafiltration (UF) membranes [144]. In addition to using the NF process in the pre-fermentation stage of winemaking to produce low alcohol wine by reducing must sugar content [74,75,76,77]; it can also be used directly in the treatment of finished wine to produce low alcohol wines [33,137,138,139,140,141,142]. Banvolgyi et al. [137] investigated the performance of an NF TriSep flat sheet membrane (type XN45) in the concentration of valuable red wine (12.8% v/v of ethanol) components. They obtained a retentate with an ethanol content of 9.8% v/v and an increase (about twice the initial concentrations) in the valuable components such as total acidity, total extract, sugar, sugarless extract, and volatile acidity. In addition, a small loss of aroma was observed in the retentate.

Catarino and Mendes [33] compared four NF membranes (NF99 HF, NF99, and NF97 from Alfa Laval and YMHLSP1905 from GE Osmonics) and one RO membrane (CA995PE from Alfa Laval) with similar MWCO (200 Da) for the dealcoholization of a red wine with 12% ABV under 16 bar and 30 °C. According to their results, all NF membranes exhibited a higher permeate flux (4.13 × 10^3^ to 7.10 × 10^3^ kgm^−2^ s^−1^) and adequate ethanol rejection (7.1% to 10.3%), resulting in dealcoholized wine samples with more promising organoleptic properties than the RO membrane. Labanda et al. [140] found similar results in their study on the reduction of the alcohol content of a white model wine with NF and RO membranes. The advantage of NF is that it preserves the organoleptic characteristics of the wine resulting from fermentation. In addition, lower transmembrane pressure is required, which makes the process more cost-effective. Moreover, the alcohol retention of NF membranes can be almost 0% [137]. Figure 2 shows the schematic of the different NF processes for wine dealcoholization.

#### 2.3.2. Reverse osmosis (RO)

RO is a membrane-based separation process that uses a hydrophilic semi-permeable membrane such as a spiral wound (SWM), a hollow fiber, a plate, and a frame or a tubular module to create a concentration or pressure gradient known as osmotic pressure between two solutions [28,182,183]. To restore equilibrium, water normally moves by osmosis through the membrane from the low concentration solution to the with high concentration solution (Figure 2). However, if enough pressure is applied to the high concentration side, a phenomenon known as reverse osmosis can occur in which the solvent is transferred from the high concentration solution across the membrane to the low concentration solution [28].

RO has been used in the beverage industry for the partial or complete dealcoholization of alcoholic wines, ciders, and beers, depending on the operating mechanism [22,140,143,145,150,184,185,186]. In one study, Catarino and Mendes [33] used a cellulose acetate RO membrane (CA995PE, from Alfa Laval) and different polyamide NF membranes with a molecular weight cut-off (MWCO) of 200 Da to produce a beverage with an alcohol content of approximately 7% to 8% v/v in a diafiltration mode. The RO membrane exhibited the lowest permeate flux and alcohol rejection, which is beneficial for the dealcoholization process. Similarly, Labanda et al. [140] studied the removal of ethanol from a model white wine using RO and NF membranes and found lower ethanol rejection and high aroma compounds rejections of during the process.

According to Bui et al. [34], reduced alcohol wines produced using RO generally have the same taste and aromas as normal wines obtained using other methods such as distillation, spinning cone technology, or arrested fermentation. When RO was used to achieve a 75% v/v ethanol reduction in an apple cider (with an original ethanol content of 8% v/v) at a pressure of 1.0 MP to 5.0 MP and at 15 °C, no significant losses of the main aroma compounds were observed [143]. This is most likely due to the lower processing temperature and better separation of the compounds in the RO process [187,188]. Contrarily, Meillon et al. [189] later reported a negative effect on odor sensations of Merlot and Shiraz wines dealcoholized by RO technique. The advantage of RO is that it requires less energy than distillation processes and that the feed wine or beverage can be processed at low temperatures of 1 °C to 5 °C. Additionally, the RO process satisfies the requirement for a “clean” technology as it allows the recovery and subsequent use of the ethanol obtained during the dealcoholization process [29]. However, a major drawback of this technique is the addition of water to achieve efficient dealcoholization, as the addition of water to wine is usually illegal or restricted in many wine-producing countries [27,72].

#### 2.3.3. Osmotic Distillation (OD) or Evaporative Perstraction (EP)

Osmotic distillation (OD), also known as evaporative perstraction (EP) or isothermal membrane distillation, is a membrane-based technology in which two aqueous phases, wine (containing the volatile compounds) and water (acting as the stripping liquid), are circulated in counter-flow on opposite sides of a hydrophobic hollow fiber membrane module (Figure 2). The driving force in this process is the partial pressure or vapor pressure of the volatile solute in wine and stripping liquid [190]. The ethanol removal mechanism in the OD process occurs in a sequence in which the ethanol in the beverage is first evaporated on the feed side of the membrane pores. Following that, the ethanol vapor diffuses through the membrane pores, exits the membrane pores, and condenses in the stripping water solution [190].

The main benefits of the OD process include low energy consumption, no thermal damage to the wine components (because the process is carried out at temperatures ranging from 10 °C to 20 °C; room temperature), slight loss of aroma and flavor, and it also meets the requirement for “clean” technology because it uses water as a stripping agent. Regarding dealcoholization, OD has demonstrated promising results for the production of low and alcohol-free beverages [35,36,147,148,149,151,190,191,192,193,194,195].

When Hogan et al. [193] used OD to remove alcohol from fermented beverages, they observed a decrease in alcohol level of up to 6% v/v at 10 °C to 20 °C with no significant losses in aroma compounds. Diban et al. [148] also investigated the partial dealcoholization of Garnacha, Xarelo, and Tempranillo wines using an industrial scale hollow fiber contactor with a membrane area of 20 m^2^, feed and stripping flow rates ranging from 600 L h^−1^ to 300 L h^−1^, a feed/stripping volume ratio ranging from 1.5:1 to 1:4.7, and pH ranging from 7 to 3. They discovered that the degree of dealcoholization increased from 1.3% to 5% v/v with minimal loss of aroma compounds (approximately 20%). Furthermore, Corona et al. [35] investigated the effect of different levels of alcohol reduction (−4.9%, −6.3%, −7.8%, −9.2%, and −10.5% v/v) on volatile compounds, phenols, and the sensory quality of *Montepulciano d’Abruzzo* red wine with an ethanol content of 13.23% v/v. From their results, OD successfully produced dealcoholized wines with satisfactory aroma profile since the dealcoholized wines 8.3% v/v (−4.9%) and 6.9% v/v (−6.3%) retained appreciable amounts of esters (above 84% and 82%, respectively), while other physicochemical parameters such as color and flavor remained unchanged in all dealcoholized wines. In contrast to other studies, a significant decrease in volatile compounds was observed at different alcohol contents of dealcoholized red wines by OD [32,36,151,196,197].

#### 2.3.4. Pervaporation (PV)

PV is a concentration-driven, membrane-based process that operates under the principle of the partial evaporation of mixtures of liquids with similar boiling points confined in an azeotropic mixture, where the liquid phase changes to the vapor phase [198,199,200,201,202]. A typical scheme of a PV process for producing reduced alcohol wines and beverages is shown in Figure 2. The transport mechanism of components via the dense polymeric membranes in a PV process can best be described by a solution-diffusion model which involves: (1) the adsorption of the target constituent from the mixture to the selective layer of the PV membrane based on its chemical affinity, (2) the diffusion of the target constituent over the membrane due to the concentration gradient, and (3) the desorption of the component at the permeate side of the membrane [203,204,205,206,207].

The PV technique has been used to remove ethanol and recover aroma from alcoholic wines and beverages [29,153,157,199,203,208,209]. For example, an experimental pilot-scale PV plant was successfully used to produce both alcohol-free and low alcoholic wines [29]. Sun et al. [210] also investigated the production of alcohol-free wine and grape spirits using PV membrane technology and discovered that the membrane technology produced more aroma compounds, resulting in a wine with a better smell and taste than traditional distilled liquor. PV has several benefits, including (1) low energy consumption and the absence of extra solvents [200], (2) high selectivity, low operation temperature, and lower loss of aroma compounds [211], and (3) a “clean” technology because less waste is produced [210]. However, there are some drawbacks to using PV, such as low permeation rates at low temperatures and a limited membrane market.

#### 2.3.5. Vacuum Distillation or Distillation under Vacuum (VD)

Vacuum distillation (VD) is a heat or thermal process that involves evaporation, distillation, and condensation, all of which occur under vacuum conditions [162,164]. VD is a well-known technique used for must self-enrichment and wine alcohol content adjustment [159,160,161,212], with many wineries already equipped for its operation. It can also be used to extract nearly all of the alcohol from a wine sample or separate it from the less volatile components. Furthermore, at the end of the treatment, the first part of the distillate can be recovered and added to the dealcoholized portion.

VD can reduce the ethanol evaporation temperature to 15–20 °C [160] and operate at pressures less than 0.1 bar to produce alcohol-free wines containing less than 1% v/v ethanol [159]. Figure 2 presents a schematic diagram of a VD process for producing reduced alcohol wines and beverages. The low operating temperature and recovery of the first portions of the distillate, which are rich in aromatic compounds, can help to reduce volatile compound losses. Taran et al. [161] investigated the effect of pressure (−0.6, −0.7, −0.8, −0.9, and −1 kgf/cm^2^) on the physicochemical properties of white and red wines dealcoholized using VD. Their findings revealed a rapid decrease in alcohol content of 2.9% v/v for red wine and 2.2% v/v for white wine at a pressure of −1 kgf/cm^2^. In comparison to membrane contactor technology, VD causes a greater reduction in alcohol and a greater increase in the concentration of other compounds such as flavonoids, organic acids, anthocyanins, and cations [160]. Despite the ability of VD technology to reduce ethanol, it can cause losses of nearly all volatile compounds, primarily ethyl esters and aliphatic alcohols [164].

#### 2.3.6. Spinning Cone Column (SCC)

The SCC operates at high speeds and low temperatures, and it is regarded as a highly effective and cost-effective method in the food industry for retaining and conserving volatile aroma compounds from slurries or liquids [28,213,214]. SCC has been used in the wine industry to produce grape must concentrate, remove sulfur dioxide from grape must, recover volatile aroma compounds, and reduce ethanol concentrations in wines [5,28,162,163,164]. The SCC is composed of a rotating vertical shaft and vertically packed cones that rotate alternately and are fixed, making it a type of falling film contactor [28,213,214], as shown in Figure 2. The SCC method involves a two-stage process for lowering the alcohol content of finished wines. The wine is passed through the SCC in the first stage at a low vacuum pressure (0.04 atm) and temperature (around 28 °C) to recover volatile wine aromas in approximately 1% of total product volume, while in the second stage the wine is dearomatized at a slightly higher vacuum pressure and temperature of around 38 °C to remove the alcohol. The dealcoholized wine is then made by combining the recovered wine aromas with the dealcoholized and dearomatized portion [28,163].

The continuous removal of ethanol from a fermenting yeast broth was studied using the SCC method which removed a large amount of ethanol (85%). However, it was discovered that the vacuum applied to the SCC affected the productivity of the yeast cells, causing the cells to shrink in size and taking on a different shape [215]. Belisario-Sánchez et al. [163] investigated the effect of SCC dealcoholization on the phenolic composition of 19 wine samples (13 red, 2 rose, and 4 white). They discovered significant differences in the total phenolics, resveratrol, tartaric esters, flavonols, and free radical-scavenging activities between the original wines and dealcoholized wines, which they attributed to changes in sulfur dioxide concentrations (which were removed with the alcohol) and volumetric variations that resulted in the amounts of components in the reduced alcohol. In addition, 2% v/v ethanol was successfully removed from a Chardonnay grape must midway through fermentation with SCC. Nonetheless, the content of fusel alcohols and volatile compounds reduced by 45% and 25%, respectively [216]. To improve the aroma recovery in red wine (cv. Tempranillo), rose wine (cv. Cabernet Sauvignon), and white wine (cv. Chardonnay) during dealcoholization by SCC distillation, Belisario-Sánchez et al. [162] manipulated the base wine aromatic extraction percentage and flow rate and recovered 97% to 100% of the total aroma fraction of all wines. The major disadvantages of SCC include the high cost of the equipment and the cost associated with its operation [166].

#### 2.3.7. Multi-stage Membrane-based Systems

Recent studies have reported the successful use of multi-stage membrane-based systems (Figure 3) for the dealcoholization of alcoholic beverages (wine and beer) to overcome many of the problems (e.g., loss of desirable volatile aroma compounds) associated with single membrane dealcoholization methods such as NF, RO, and OD [80,144,146,168,169,170,217]. For the production of low-alcohol beer, a multi-stage membrane-based process consisting of PV and distillation units was used, and aroma compounds were recovered and incorporated into the dealcoholized beer, resulting in a non-alcoholic beer (0.5% v/v ethanol) with a good flavor profile [158]. Salgado et al. [80] investigated the performance of a combined nanofiltration-pervaporation (NF-PV) system at a pilot-scale in the production of a full-flavored low-alcohol white wine. After reducing the must sugar content with a two-stage NF, PV was used to recover aroma precursors from the must. Finally, the recovered aroma precursors were reconstituted with the reduced sugar must from the two-stage NF, and then fermented into a low alcohol wine. They found that combining musts from NF and volatile aromas from PV had a similar aroma content to the original must and that combining the NF and PV processes produced the best wine.

Pham et al. [169] recently investigated the effect of partial dealcoholization on the composition and sensory properties of Cabernet Sauvignon wines using a reverse osmosis-evaporative perstraction (RO-EP) technique, after which they created a reconstituted wine from 50:50 blends of control and dealcoholized wines and further characterized the sensory consequences of partial dealcoholization. They found no significant differences between the reconstituted wine and the control wine in terms of the overall aroma intensity. The RO-EP technique usually consumes low energy and operates at ambient temperature and atmospheric pressure, hence causing low thermal damage to wine components [190]. Nonetheless, it has the potential to significantly reduce desirable odor-active volatiles such as ethyl acetates, esters, and monoterpenes [151]. Despite the ability of multi-stage processes to better preserve wine aroma and taste within certain limits, losses of important aroma compounds still occur. Further aroma enhancement after dealcoholization via multi-stage processes could still help wine producers in limiting undesirable effects in the wine and increasing acceptability amongst consumers.

## 3. Impact of Dealcoholization Techniques on Wine Quality 

### 3.1. Impact on phenolic composition

The phenolic composition of wine is made up of flavonoids and non-flavonoids [218]. Flavonoids include flavones, flavanols ((+)-catechin and (−)-epicatechin), flavonols (quercetin, myricetin, kaempferol, and rutin), anthocyanins, and proanthocyanidins while non-flavonoids are mainly resveratrol (3,4,5-trihydroxystilbene), hydroxybenzoic acids (p-hydroxybenzoic, vanillic, syringic, gallic, gentisic, salicylic, and protocatechuic acids), and hydroxycinnamic acids (caffeic, coumaric, and ferulic acids) [39,219,220,221,222,223]. Regarding wine quality, especially red wine, phenolic compounds play a vital role by contributing to organoleptic properties such as astringency and color [224]. Health-wise, phenolic compounds can be effective in the prevention of cardiovascular diseases [225,226,227]. Although changes in alcohol content do not generally affect basic wine parameters such as density, pH, titratable acidity, and volatile acidity [168,228], these changes have been reported to influence wine phenolic compounds [150,163,168,189]. Important findings from some studies on the phenolic composition of wines dealcoholized by physical dealcoholization methods are summarized in Table 2.

The dealcoholization of white, rose, and red wines by SCC distillation at pilot plant scale was reported to cause minimal damage to phenolic compounds such as flavonols, tartaric esters, stilbenes (specifically *trans*- and *cis*- resveratrol), flavonols (i.e., rutin, quercetin, and myricetin), flavan-3-ols (mainly (+)-catechin and (−)-epicatechin), anthocyanins (in particular malvidin 3-glucoside), and non-flavonoids (including gallic, caffeic, and p-coumaric acids) [163]. Additionally, the technique increased the concentrations of these compounds in the wines after dealcoholization [163]. Phenolic compounds such as polyphenols and anthocyanins were not lost during the dealcoholization (at 5% v/v ethanol) of Rosé, Pelaverga, and Barbera red wines using a membrane contactor and VD method [134]. Recently, Liguori et al. [36] studied the main quality parameters of white wine (cv Falanghina, 12.5% v/v) dealcoholized at different ethanol concentration levels ranging from 9.8% to 0.3% by an osmotic distillation process. There were no significant differences in flavonoids, total phenols, total acidity, and organic acids between the wine samples at different alcohol content levels. Similar results were obtained in a red wine dealcoholized at different alcohol levels [35]. Furthermore, when RO-EP treatment was used in the partial dealcoholization (i.e., a reduction of 0.5% to 5.0% ABV) of red wine, it resulted in increased phenolics, color intensity, and organic acids [168]. In contrast, a significant change in the color of red wines dealcoholized by RO was observed [229]. The increase in phenolic compounds in wine, particularly anthocyanins, after dealcoholization noted in most of these studies may be due to reduced precipitation of wine tartrate salts [22], as wine tartrate salts can absorb polyphenols [230]. It has also been reported that dealcoholization at a low temperature (20 °C) can lead to higher retention of polyphenols in wine [138]. In addition, the increment can be attributted to the concentration effect produced by the removal of ethanol from the wine [163].

### 3.2. Impact on Volatile Composition

The composition of volatile compounds influences the overall aroma and flavor of wine [231,232,233,234,235]. Wine contains over 1000 volatile compounds of various chemical classes (alcohols, esters, fatty acids, aldehydes, terpenes, ketones, and sulfur compounds), and wine fermentation produces approximately 400 volatile compounds [236]. During dealcoholization, the removal of alcohol from wine is usually accompanied by the removal of water and some volatile compounds as well [27]. Table 3 summarizes some findings regarding the volatile composition of wines during the dealcoholization process. In the case of membrane contactor techniques such as RO, NF, PV, and OD that use a membrane for ethanol removal, a greater pressure difference across the membrane than the osmotic pressure difference causes ethanol and water from the wine to pass through the membrane [28].

Several studies have reported on the use of membrane techniques in wine dealcoholization and their subsequent effect on the dealcoholized wine volatile compositions [29,32,35,36,140,143,146,151,160,190]. A low alcohol content apple cider was produced by RO with a polyamide membrane AFC99 in both batch and diafiltration configurations [143]. The process was operated at 15 °C and 45 bar with a feed flow of 200 L h^−1^. During the batch configuration process, 50% of ethanol was removed with an estimated loss of 77% of total higher alcohols, 20% of total aldehydes, 25% of total acids, and 25% of total esters. In the diafiltration configuration, estimated losses of 96% total higher alcohols, 43% total aldehydes, 18.5% total acids, and 28% total esters accompanied the removal of 75% ethanol. However, losses in these volatile compounds were deemed insignificant in both configurations [143]. Takács et al. [29] used PV in the total dealcoholization of a Tokaji Hárslevelű wine (13.11% v/v), resulting in a 70% loss of the total aroma conpounds, but the loss of individual aroma compounds was not reported. When Sun et al. [210] used PV technology to reduce the alcohol content of a Cabernet Sauvignon red wine from 12.5% to 0.5%, they discovered losses of volatile compounds, specifically alcohols (40%), acids (28%), and esters (99%). After dealcoholization with a polyvinylidene fluoride membrane, Varavuth et al. [190] found losses of 47% to 70% and 23% to 44% of ethyl acetate and isoamyl alcohol, respectively, in a model wine solution. Diban et al. [147] used the same polyvinylidene fluoride membrane to measure the losses of eight volatile compounds in wine and wine model solution after a 2% v/v ethanol reduction, but only losses were observed in model solution after a 5% v/v ethanol reduction. Furthermore, Belisario-Sánchez et al. [162] found that after dealcoholization by SCC, the total volatile aroma compounds of Tempranillo red wine, Cabernet Sauvignon rose wine, and Chardonnay white wine were lost by approximately 18%, 4%, and 9%, respectively.

During dealcoholization, volatile compounds are lost in the same way as ethanol. As a result, their original contents are lost during dealcoholization due to vaporization and diffusion [32,190]. In addition, some losses of 2% to 3% have been attributed to their adsorption onto the membrane [147]. This is due to their high affinity for the membrane and high volatility, which allows them to pass through the membrane more easily. Through a non-covalent interaction between the polyphenols and the aromatic ring of aromatic compounds, the non-volatile matrix of wine, particularly polyphenols, can also aid in the stability and retention of volatile compounds [32]. This best explains why a 50% reduction in the ethanol content of a 13% v/v Aglianico wine by a membrane contactor technique did not affect the amount of 2-phenylethanol in the dealcoholized wine [151]. However, when higher ethanol concentrations were removed, a drastic decrease in the 2-phenylethanol concentration was observed, which was attributed to weaker 𝜋–𝜋 stacking caused by the decrease in ethanol content (7% v/v) of the wine.

The operating conditions used during the dealcoholization process can also have an impact on the concentrations of wine volatile compounds. A change in some operating conditions of an OD process, such as lowering the temperature from 20 °C to 10 °C and changing the positions of the feed and stripping streams from a previous study [151], helped to decrease the loss of volatile aroma compounds by about 2.8% during the dealcoholization of a 12.5% v/v white wine [36]. From the findings, it is evident that the physical technologies used in the dealcoholization of wines can result in significant losses of volatile compounds due to the reduction in alcohol levels. However, the significance and extent of the changes can also depend on the operating conditions applied, the type of membrane used, and the non-volatile matrix of the wine.

### 3.3. Impact on Sensory Characteristics

Ethanol is the most abundant of the volatile compounds in wine and its concentration can influence the perception of wine aroma and flavor as well as several mouthfeel and taste sensations [147,228,237,238]. Higher ethanol concentrations in wine typically enhance sensitivity to body, bitterness, and hotness, whereas lower concentrations can reduce the perception to aroma, flavor, acidity, and astringency [19,20,239,240,241]. Some studies have been conducted to investigate the sensory quality of wines or wine model solutions during ethanol removal [29,140,147,150,163,164,189,190,197]. The sensory profile of wine after partial or total dealcoholization is primarily determined by the amount of alcohol remaining in the dealcoholized wine [28,196,242,243]. Table 4 summarizes the key findings from some of these studies on the sensory changes caused by dealcoholization.

The loss of esters from 11% to 100% as a result of alcohol reduction at −2, −3, −5% v/v by a polypropylene hollow fiber membrane contactor apparatus in two red wines (cv. Aglianico) with different initial alcohol contents (15.37% and 13.28% v/v) resulted in a decrease of cherry and red fruits olfactory notes [32]. Furthermore, at 5% v/v dealcoholization, a significant increase in astringency and acidity in both wines was observed [32]. Meillon et al. [189] found a decrease in red fruits and blackcurrant odors after a dealcoholization (−1.5% and −3%) of a Merlot wine by RO, whereas the red fruits note was found to decrease only after a reduction of 3 alcoholic degrees in Syrah wine [189]. However, after the ethanol reduction of Merlot and Syrah wines by RO, the prevalence of bitter taste decreased [189]. Using the temporal dominance of sensations (TDS), the sensation of prevalent berries was found to decrease when the alcohol content of wine was lower than 9.5% v/v, specifically at −4% and −5.5% after the dealcoholization of Syrah wine by RO [244]. When the ethanol content of oaked Chardonnay wine was gradually reduced using SCC technology, the sensations of hot mouthfeel and overall aroma intensity were significantly reduced when compared to the original wine [245]. A decrease in the bitterness and hotness in wines with low alcohol is expected as it is evident that ethanol causes apparent hotness and bitterness [189,242,245,246,247,248].

The use of OD technology to partially dealcoholize red wine was reported to increase astringency [32,150]. Except for wines with high tannin concentrations such as Cabernet Sauvignon, ethanol can mask astringency in both wine and model solutions [224,238]. A dealcoholized white wine (cv Falanghina, 0.3% vol) had high acidity, low odor, sweetness, and body in contrast to its original wine (cv Falanghina, 12.5% vol), making it unbalanced in taste with an unpleasant aftertaste [36]. When two Aglianico wines with different alcohol levels were partially dealcoholized at 2% v/v ethanol using a membrane contactor technique, the higher-alcohol wine (15.4% v/v) showed more reductions in red fruit and cherry sensations, while the lower-alcohol wine (13.3% v/v) was perceived as more floral [245]. This is most likely due to the polarity of the lower-alcohol wine, which has a greater influence than other factors such as aroma loss.

Differences in the non-volatile phenolic matrix composition of the wine, particularly the phenolic concentration, may also have contributed to these findings, as some studies have shown that phenolic compounds can influence the aroma perception of red wines [249,250]. Reduced alcohol wines are typically associated with poor sensory quality, such as lack of body, flavor imbalance, reduced heat sensation, increased astringency, increased bitterness, and high acidity when compared to original wines [189,242,243,245,246,247]. However, up to a certain reduction level, their preference over original wines is still evident [244]. For example, Lisanti et al. [32] found no major differences between the original wine and one with a −2% v/v decrease in alcohol content after dealcoholization by membrane contactor technique. On the contrary, increasing the degree of reduction by −3% and −5% v/v increased the astringency, acidity, and bitterness of the wines. Similarly, a reduction in alcohol by 2% and 4% v/v in a 13.4% v/v Syrah wine using RO did not significantly affect the liking of the partially dealcoholized wines. However, a reduction of 5.5% v/v significantly decreased the preference [244]. Furthermore, when Corona et al. [35] dealcoholized (−8% v/v) a red wine (cv. Montepulciano d’Abruzzo) with an initial alcohol content of 13.2% v/v, they found no significant differences in the color intensity and overall acceptability between the two wines. In a white Chardonnay wine (14.2% v/v), a similar pattern was also found, with no major variations observed when wines were reduced by 1.5% and 3.3% v/v, while a reduction above 4.5% v/v negatively influenced consumer liking. These findings could be attributed to most consumers’ failure to perceive alcohol reductions of less than 2% v/v [27].

## 4. Conclusions and Potential Challenges

The removal or reduction of alcohol content of wine and other alcoholic beverages has been the interest of some winemakers and researchers over the past years as trends in wine styles as well as climate change has affected the consumption of alcoholic wines. To achieve this, different technologies have been used at the various stages of winemaking. Although the technologies used during the pre-fermentation and fermentation stages show promising results, post-fermentation alcohol reduction techniques, particularly membrane separation (nanofiltration, reverse osmosis, evaporative perstraction, and pervaporation) and thermal distillation (vacuum distillation and spinning cone column) represent the most common and widely used commercial techniques in the production of dealcoholized beverages. Despite the ability of these techniques to preserve the phenolic components, volatile composition, and sensory qualities of wine within certain limits of dealcoholization, problems such as changes in color and losses of desirable volatile aroma compounds, which subsequently affect the sensory quality could occur.

Additionally, the operational costs related to these techniques are relatively high. Moreover, reduced-alcohol or alcohol-free wines and beverages may be susceptible to microbial contamination and should be produced under aseptic conditions. Despite these challenges, many commercial reduced alcoholic strength products have been produced and marketed successfully. Moreover, the combination of some of the techniques as well as reconstitution (aroma enhancement) after dealcoholization may provide a good alternative for balancing production costs and the sensory profile of reduced-alcohol or alcohol-free wines and beverages.

A greater understanding of the various post-fermentation dealcoholization techniques and their influence on wine quality during the dealcoholization process as well as an understanding of growing consumer trends and the non-alcoholic wine market will help winemakers in choosing the best technique to limit adverse effects and help meet the needs and acceptance amongst differently targeted consumers such as younger people, pregnant women, drivers, and teetotalers.

## Figures and Tables

**Figure 1 foods-10-02498-f001:**
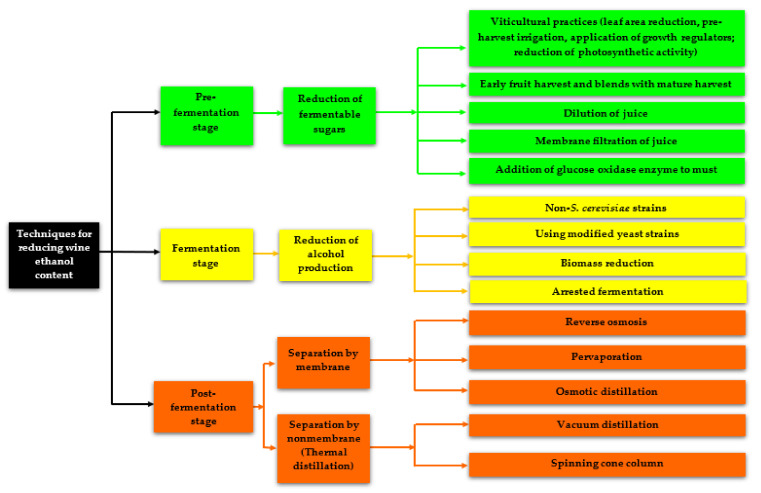
Techniques for alcohol reduction in wines and fermented beverages.

**Figure 2 foods-10-02498-f002:**
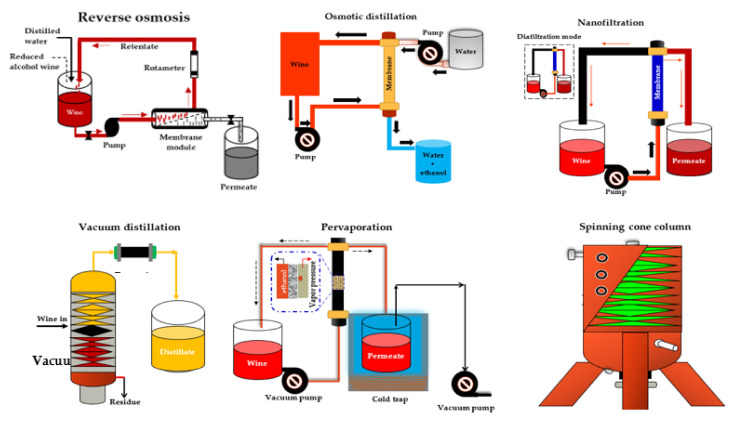
Scheme of different integrated systems for wine dealcoholization. Adapted from Salgado et al. [80], Catarino and Mendes [158], Pham et al. [168], and Liguori et al. [181].

**Figure 3 foods-10-02498-f003:**
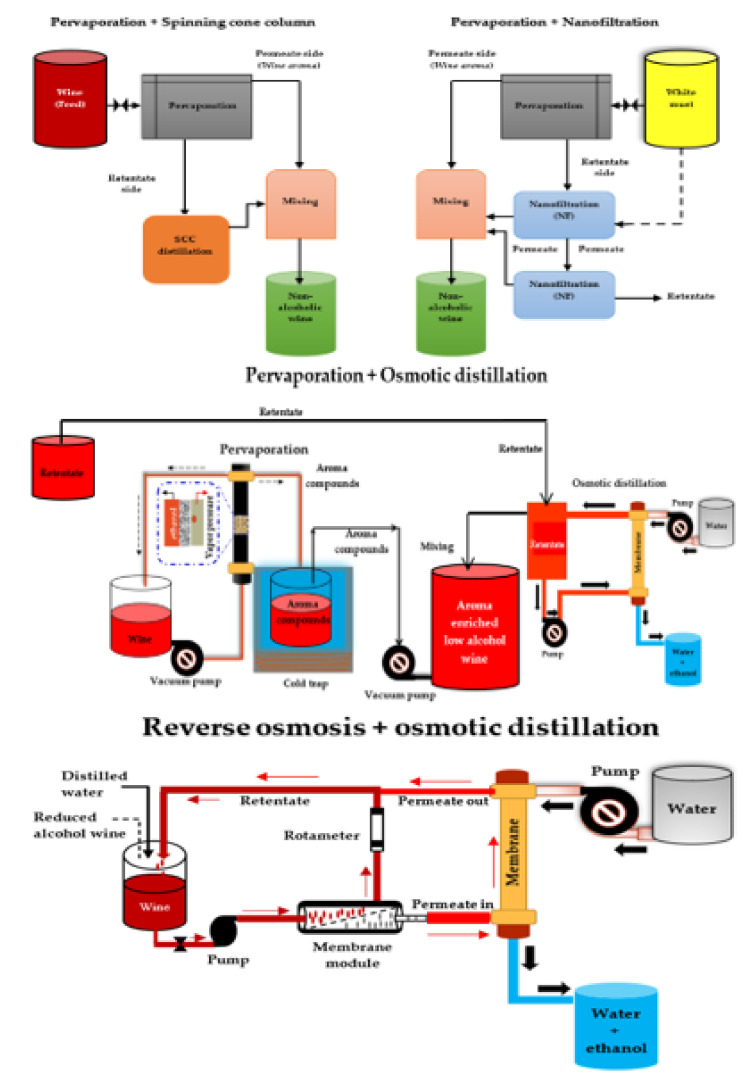
Scheme of different multi-stage membrane-based systems for wine dealcoholization. Adapted from García-Martín et al. [74], Liguori et al. [152], and Belisario-Sánchez et al. [162].

**Table 1 foods-10-02498-t001:** Different techniques to reduce wine alcohol content in the several stages of wine production.

Stage of Wine Production	Ethanol Removal Process	Technology	Alcohol Content Reduction	References
Pre-fermentation	Reduction of fermentable sugars	Viticultural practices (leaf area reduction, pre-harvest irrigation, application of growth regulators; reduction in photosynthetic activity)	Up to 2% v/v	[37,38,39,40,41,42,43,44,45,46,47,48,49,50,51,52,53,54,55]
		Early fruit harvest and blends with mature harvest	Up to 3% v/v	[56,57,58,59,60,61,62,63,64,65,66]
		Dilution of grape must	Up to 7% v/v	[61,62,63,67,68,69,70,71,72,73]
		Filtration of must	Up to 5% v/v	[74,75,76,77,78,79,80,81,82,83]
		Addition of enzyme (glucose oxidase)	Up to 4% v/v	[5,84,85,86,87,88,89]
Fermentation	Reduction of alcohol production	Use of Non-*Saccharomyces cerevisiae* yeasts	Up to 2% v/v	[90,91,92,93,94,95,96,97,98,99,100,101,102,103,104,105,106,107,108,109,110,111,112,113,114,115,116,117,118,119,120]
		Use of modified yeast strains	Up to 3.6% v/v	[121,122,123,124,125,126,127,128,129,130,131,132]
		Biomass reduction	Up to 4% v/v	[133,134,135]
		Arrested fermentation	High reduction	[5,136]
Post-fermentation	Separation by membrane	Nanofiltration (NF)	Up to 4% v/v	[77,137,138,139,140,141,142]
		Reverse osmosis (RO)	Up to 0.5% v/v or less	[22,32,34,140,143,144,145,146]
		Osmotic distillation (OD)	Up to 0.5% v/v or less	[32,35,36,146,147,148,149,150,151,152]
		Pervaporation (PV)	Up to 0.5% v/v or less	[29,153,154,155,156,157,158]
		Vacuum distillation (VD)	Up to 1% v/v or less	[159,160,161]
		Spinning cone column (SCC)	Up to 0.3% v/v	[28,162,163,164,165,166]
		Multi-stage membrane-based systems	Up to 0.5% v/v or less	[80,144,167,168,169,170]

**Table 2 foods-10-02498-t002:** Some reported changes in wine phenolic compounds using different dealcoholization processes.

Wine Type	Dealcoholization Process	Alcohol Reduction		Reported Effects on Phenolic Composition	Reference
		Co (% v/v)	Cf (% v/v)		
Red wine	NF	12.0	6.0–4.0	Reduction in wine alcohol volume by a factor of 4 leads to 2.5–3 times more anthocyanins and resveratrol in the wine concentrates	[138]
Cabernet Sauvignon–Merlot–Tempranillo red wine	RO	12.7	4.0–2.0	No significant differences were observed in total anthocyanins and phenolic compounds for both original and dealcoholized wines. Colour intensity increased by around 20% in dealcoholized wines (due to the concentration effect from the removal of ethanol as well as the retention of anthocyanins by the membrane), while the tonality diminished by around 15%	[229]
Cabernet Sauvignon red wine	RO	14.8	13.8–12.8	The total phenolic index, total proanthocyanidins, and percentages of procyanidins, prodelphinidins, and galloylation of partially dealcoholized wines and the control wine remains almost unchanged and did not differ. Control wine and partially dealcoholized wines have statistically similar total anthocyanin concentrations with no observed color differences between these wines	[22]
Grenache–Carignan red wine	RO	16.2	15.1–14.1	The total phenolic index and total proanthocyanidins of partially dealcoholized wines and the control wine remain almost unchanged and do not differ. Slight but statistically significant differences were observed in the percentages of procyanidins, prodelphinidins, and galloylation during alcohol reduction. Total anthocyanin concentrations of partially dealcoholized wines were statistically significantly higher than that of the control wine	[22]
Montepulciano d’Abruzzo red wine	RO	13.2	9.0	Increase in total phenols and decrease in total anthocyanins during ethanol reduction in wine samples. Color intensity increases during ethanol removal	[146]
Aglianico red wine	OD/EP	12.8	4.9–0.4	Higher amount of total phenols in dealcoholized wine samples compared to the original wine. Color intensity decreased slightly at the end of dealcoholization	[197]
Aglianico red wine	OD/EP	15.4	13.5–10.8	The alcohol removal process did not affect the content of vanillin reactive flavans and total phenolics. A loss of 49% of total monomeric anthocyanins was observed after dealcoholization while total anthocyanins remained almost unchanged with no significant differences. Color parameters of dealcoholized wines were not significantly different compared to the original wine after alcohol removal	[150]
Merlot red wine	OD/EP	13.8	11.1–8.9	The alcohol removal process did not affect the content of vanillin reactive flavans and total phenolics. A loss of 57% of total monomeric anthocyanins was observed after dealcoholization while total anthocyanins remained almost unchanged with no significant differences. Color parameters of dealcoholized wines were not significantly different compared to the original wine after alcohol removal	[150]
Piedirosso red wine	OD/EP	13.6	11.5– 8.4	The alcohol removal process did not affect the content of vanillin reactive flavans and total phenolics. A loss of 52% of total monomeric anthocyanins was observed after dealcoholization while total anthocyanins remained almost unchanged with no significant differences. Color parameters of dealcoholized wines were not significantly different compared to the original wine after alcohol removal	[150]
Aglianico red wine	OD/EP	12.5	10.6	No significant differences between base wine and dealcoholized wine in terms of total polyphenols and color intensity	[152]
Barbera red wine	OD/EP	15.2	5.0	Higher contents of total anthocyanins and total flavonoids compared to the original wine. Color: the intensity increases and the hue decreases (loss of orange notes) due to the increased content of total anthocyanins	[160]
Langhe Rosè wine	OD/EP	13.2	5.0	Higher contents of total anthocyanins and total flavonoids compared to the original wine. Color: the intensity increases and the hue decreases (loss of orange notes) due to the increased content of total anthocyanins	[160]
Verduno Pelaverga red wine	OD/EP	14.6	5.0	Higher contents of total anthocyanins and total flavonoids compared to the original wine. Color: the intensity increases and the hue decreases (loss of orange notes) due to the increased content of total anthocyanins	[160]
Falanghina white wine	OD/EP	12.5	9.8–0.3	At different alcohol content levels of wines, the total phenols and flavonoids do not differ significantly as they remain almost unchanged during the alcohol removal process	[36]
Montepulciano d’Abruzzo red wine	OD/EP	13.2	8.3–5.4	Both total phenols and total anthocyanins decrease in dealcoholized wines with no significant differences compared to the original wine. The color intensity remains almost unchanged during ethanol removal	[146]
Montepulciano d’Abruzzo red wine	OD/EP	13.2	8.3–2.7	Flavonoids and phenolic compounds remain almost unchanged in all dealcoholized samples compared to the base wine with no significant differences. Color intensity (evaluated by flavonoids and phenolic compounds) decrease slightly in all dealcoholized samples	[35]
Langhe Rosè wine	VD	13.2	5.0	Higher contents of total anthocyanins and total flavonoids compared to the original wine. Color the intensity increases and the hue decreases (loss of orange notes) due to the increased content of total anthocyanins	[160]
Barbera red wine	VD	15.2	5.0	Higher contents of total anthocyanins and total flavonoids compared to the original wine. Color: the intensity increases and the hue decreases (loss of orange notes) due to the increased content of total anthocyanins	[160]
Verduno Pelaverga red wine	VD	14.6	5.0	Higher contents of total anthocyanins and total flavonoids compared to the original wine. Color the intensity increases and the hue decreases (loss of orange notes) due to the increased content of total anthocyanins	[160]
Red wine	SCC	14.0	< 0.3	Increase in phenolic compounds, total phenolic, flavonol, tartaric ester, and anthocyanin contents by approximately 24%. Higher content of resveratrol than the original wine	[163]
Rose wine	SCC	14.0	< 0.3	Increase in phenolic compounds, total phenolic, flavonol, tartaric ester, and anthocyanin contents by approximately 24%. Higher content of resveratrol than the original wine	[163]
White wine	SCC	14.0	< 0.3	Increase in phenolic compounds content by approximately 24%	[163]
Montepulciano d’Abruzzo red wine (cv.)	RO–OD/EP	13.2	7.1–5.5	Total phenols increase while total anthocyanins decrease in the dealcoholized wine samples. Color intensity increases during ethanol removal	[146]
Cabernet Sauvignon red wine	RO–OD/EP	14.1	12.5	Significantly increase in color intensity due to increased content of anthocyanins during alcohol reduction compared to the base wine	[168]
Shiraz red wine	RO–OD/EP	15.2	12.6	Increase in color intensity due to increased content of anthocyanins during alcohol reduction compared to the base wine	[168]

Co = original alcohol content; Cf = final alcohol content; NF = nanofiltration; RO = reverse osmosis; OD = osmotic distillation; EP = evaporative perstraction; VD = vacuum distillation; SCC = spinning cone column.

**Table 3 foods-10-02498-t003:** Some reported changes in wine volatile compounds using different dealcoholization processes.

Dealcoholization Process	Wine Type	Membrane	Operating Mode/Conditions	Alcohol Reduction		Volatile Composition		Sampling and Analytical Method	Reference
				Co (% v/v)	Cf (% v/v)	Volatile Compounds	Estimated Average Losses (%)		
NF	White model wine	TORAY–UB70	Batch retentate–recycling mode T = 15 P = 10	12.0	8.4	Diethyl succinate 2–phenyl–ethanol *cis*–3–hexenol Isovaleric acid	2.4 2.9 12.6 11.7	HS/SPME–GC/MS	[140]
	Red Wine	Polyamide, NF9, Alfa Laval	T = 30 P = 16	12.0	9.1	Total volatile aroma**	30.0	GC–FID	[33]
RO	Model wine	Osmonics–SE	Batch retentate–recycling mode T = 15 P = 17–29	12.0	8.4	Diethyl succinate 2–phenyl–ethanol *cis*–3–hexenol Isovaleric acid	0.6–1.6 2.5–3.5 7.8–11 11.9–18.1	HS/SPME–GC/MS	[140]
	Red Wine	Cellulose acetate, CA995PE	T = 30◦C P = 16	12.0	8.4	Total aroma**	90.0	GC–FID	[33]
	Montepulciano d’Abruzzo red wine	RO membrane (100 DA)	T = 10 P = ns Time = 40	13.2	9.0	Alcohols Acids Esters Phenols Lactones	30.0 22.0 8.0 13.0 14.0	SPME–GC/MS	[146]
OD/EP	Model wine	Polyvinylidene fluoride (PVDF) Memcor	Qf = 0.053 Qs = 0.093 T = 30 Time = 60	13.0	8.1	Isoamyl alcohol Ethyl acetate	44.0 70.0	GC–FID	[190]
	Falanghina white wine	Liqui–Cel 0.5 × 1, PP hollow fiber	Qf = 0.07 Qs = 0.14 T = 10 Time = 240	12.5	9.8–0.3	Higher alcohols Acids Esters Ketones lactones	49.5–98.9 60.5–98.7 71.5–99.0 67.1–99.9 73.6–98.2	LE–GC/MS, LE–GC/FID	[36]
	Xarelo white wine	Liqui–Cel ExtraFlow	Qf = 10 Qs = 10 T = room temperature Time = 20	11.5	10.1	Isoamyl acetate Ethyl hexanoate Ethyl octanoate Ethyl decanoate	27.0 37.0 28.0 24.0	SBSE–GC/MS	[148]
	Soave white wine	PTFE hollow fiber (Teflon, Verona, Italy)	Qf = 0.2 Qs = 0.2 T = 20 Time = ns	ns	*	Alcohols Acids Esters Terpenes	12.6–32.2 5.6–16.4 34.0–58.4 22.0–26.0	SPE–GC/MS	[196]
	Verdicchio white wine	PTFE hollow fiber (Teflon, Verona, Italy)	Qf = 0.2 Qs = 0.2 T = 20 Time = ns	ns	*	Alcohols Acids Esters Terpenes	8.9–25.8 8.0–15.8 40.0–54.1 21.0–28.0	SPE–GC/MS	[196]
	Aglianico red wine	Liqui–Cel Extra–flow, PP hollow fiber	Qf = 0.583 Qs = 0.183 T = 20 Time = 283	13.8	11.6–8.8	Alcohols Esters Acids Terpenes *Others:* Benzaldehyde 𝛾–Butyrolactone	8.4–31.8 42.9–60.9 12.5–17.1 13.8–32.3 55.3–65.9 4.5–13.6	SPE–GC/MS	[32]
	Aglianico red wine	Liqui–Cel Extra–flow, PP hollow fiber	Qf = 0.583 Qs = 0.183 T = 20 Time = 283	15.5	13.5–10.8	Alcohols Esters Acids Terpenes *Others:* Benzaldehyde 𝛾–Butyrolactone Vitispirane	9.2–13.7 33.8–50.6 11–18.5 3.6–14.5 nf 12.9 Unc	SPE–GC/MS	[32]
	Aglianico red wine	Liqui–Cel 0.5×1, PP hollow fiber	Qf = 0.07 Qs = 0.14 T = 20 Time = 255	13.0	6.5–0.2	Alcohols Acids Esters Sulfur compounds Phenols Ketones and lactones Aldehydes	57.9–99.9 23.6–78.9 12.8–89.9 2.1–78.7 66.7–100 23.6–97.9 unc–100	LE–GC/MS, LE–GC/FID	[151]
	Merlot red wine	Liqui–Cel Extra–flow, PP hollow fiber	Qf = 5.8 Qs = 8.1 T = 20 Time = 60	13.4	11.3	Ethyl acetate Isoamyl acetate Isoamyl alcohol Ethyl hexanoate Ethyl octanoate Linalool 2–Phenylethyl acetate	37.4 34.9 13.7 33.0 67.8 14.5 13.6	HS/SPME–GC/MS	[147]
	Barbera red wine	Polypropylene hollow fibers (JU.CLA.S. LTD, Verona, Italy)	Qf = 1.6 Qs = 0.8 T = 10 Time = 360	14.6	5.0	Alcohols Acids Esters	63.9 17.4 23.8	SPE–GC/FID	[160]
	Tempranillo red wine	Liqui–Cel ExtraFlow	Qf = 5.8 Qs = 5.8 T = room temperature Time = 60	13.3	9.0	Isoamyl alcohol Ethyl hexanoate	21.0 20.0	SBSE–GC/MS	[148]
	Garnacha red wine	Liqui–Cel ExtraFlow	Qf = 5 Qs = 5 T = room temperature Time = 60	13.9	9.3	Isoamyl acetate Ethyl hexanoate	24.0 36.0	SBSE–GC/MS	[148]
	Verduno Pelaverga red wine	Polypropylene hollow fibers (JU.CLA.S. LTD, Verona, Italy)	Qf = 1.6 Qs = 0.8 T = 10 Time = 360	14.6	5.0	Alcohols Acids Esters	59.9 23.6 45.2	SPE–GC/FID	[160]
	Montepulciano d’Abruzzo red wine	Liqui–Cel 0.5×1, PP hollow fiber	Recycling mode Qf = 1.5 Qs = 0.5 T = 10 Time = 240	13.2	8.3–2.7	Alcohols Acids Esters Lactones Phenols *Others:* Benzaldehyde α–Terpineol	56.0–84.0 18.0–23.0 64.0–85.0 11.0–37.0 11.0–37.0 2.0–26.0 5.0–49.0	SPE– LE–GC/MS/FID	[35]
	Montepulciano d’Abruzzo red wine	Liqui–Cel mini module 1.7x5.5 Membrana	Recycling mode Qf = 1.5 Qs = 0.5 T = 10 Time = 120	13.2	8.3–5.4	Alcohols Acids Esters Phenols Lactones	2.0–3.0 18.0–25.0 15.0–19.0 5.0–10.0 7.0–25.0	SPME–GC/MS	[146]
	Langhe Rosè wine	Polypropylene hollow fibers (JU.CLA.S. LTD, Verona, Italy)	Qf = 1.6 Qs = 0.8 T = 10 Time = 360	13.2	5.0	Alcohols Acids Esters	60.4 30.9 47.8	SPE–GC/FID	[160]
PV	Tokaji Hárslevelű white wine	PERVAP.Sulzer 1060 PDMS	‘‘Carrier gas mode’’ under atmospheric pressure T = 40–70	13.1	0.1	Total volatile aroma**	70.0	Distillation/LE–GC/MS	[29]
	Cabernet Sauvignon red wine	PDMS JS–WSM–8040 (JiuSi High–Tech, Nanjing, China)	Batch operation T = 45 VP = 0.05	12.5	0.5	Alcohols Acids Esters	19.7–39.5 12.7–28.2 48.0–99.9	GC/MS	[210]
VD	Barbera red wine	–	T = 15	15.2	5.0	Alcohols Acids Esters	50.4 13.7 19.8	SPE–GC/FID	[160]
	Verduno Pelaverga red wine	–	T = 15	14.6	5.0	Alcohols Acids Esters	53.6 2.3 19.5	SPE–GC/FID	[160]
	Langhe Rosè wine	–	T = 15	13.2	5.0	Alcohols Acids Esters	51.4 2.5 22.9	SPE–GC/FID	[160]
SCC	White wine	–	T = 25 VP = 0.08 Time = 60	10.6	0.3	Aliphatic alcohols Aromatic alcohols Acids Esters Ketones	98.0 3.0 20.0 53.0 71.0	LE–GC/FID	[164]
	Chardonnay white wine	–	T = 30 VP = 0.04 Time = 60	ns	ns	Total aroma**	1.0–9.0	HS/SPME–GC/MS	[162]
	Tempranillo red wine	–	T = 30 VP = 0.04 Time = 60	ns	ns	Total aroma**	3.0–18.0	HS/SPME–GC/MS	[162]
	Cabernet Sauvignon rose wine	–	T = 30 VP = 0.04 Time = 60	ns	ns	Total aroma**	1.0–4.0	HS/SPME–GC/MS	[162]
RO-OD/EP	Shiraz red wine	Memstar AA MEM–074 and Liqui–Cel 2.5×8 Extra–flow PP hollow fiber	Qf = ns Qs = ns T = ns P = ns Time = ns	16.3	13.3–10.4	Alcohols Esters Monoterpenes C13–Norisoprenoids Lactones Others: Dimethyl sulfide	14.9–38.9 29.8–49.5 9.2–20.8 9.4–14.5 17.1–21.4 52.6–71.9	HS–SPME–GC/MS	[217]
	Montepulciano d’Abruzzo red wine	RO membrane (100 DA) and Liqui–cel mini module 1.7×5.5 Membrane	Recycling mode Qf = 1.5 Qs = 0.5 T = 10 P = ns Time = 120	13.2	7.1–5.5	Alcohols Acids Esters Phenols Lactones	17.0–27.0 19.0–24.0 15.0–22.0 16.0–18.0 unc–14.0	SPME–GC/MS	[146]
	Barossa Valley Shiraz – Cabernet Sauvignon red wine	Spiral wound 4040 and hollow fiber perstractive membrane (VA Filtration, Nuriootpa, Australia)	Qf = ns Qs = ns T = 55 P = 30 Time = 90	14.1	12.5	Alcohols Acids Esters	15.5 10.0 5.1	SPME–GC/MS	[168]
	McLaren Vale Cabernet Sauvignon red wine	Spiral wound 4040 and hollow fiber perstractive membrane (VA Filtration, Nuriootpa, Australia)	Qf = ns Qs = ns T = 55 P = 30 Time = 90	17.1	14.5	Alcohols Acids Esters	13.6 6.1 18.8	SPME–GC/MS	[168]
	Adelaide Hills Shiraz red wine	Spiral wound 4040 and hollow fiber perstractive membrane (VA Filtration, Nuriootpa, Australia)	Qf = ns Qs = ns T = 55 P = 30 Time = 90	14.9	14.2	Alcohols Acids Esters	7.0 0.4 8.6	SPME–GC/MS	[168]
	Barossa Valley Shiraz red wine	Spiral wound 4040 and hollow fiber perstractive membrane (VA Filtration, Nuriootpa, Australia)	Qf = ns Qs = ns T = 55 P = 30 Time = 90	15.2	12.6	Alcohols Acids Esters	11.0 5.6 21.2	SPME–GC/MS	[168]
	McLaren Vale Shiraz red wine	Spiral wound 4040 and hollow fiber perstractive membrane	Qf = ns Qs = ns T = 55 P = 30 Time = 90	14.7	12.3	Alcohols Acids Esters	7.1 2.5 9.7	SPME–GC/MS	[168]
	Cabernet Sauvignon red wine A	Spiral wound 4040 and hollow fiber perstractive membrane (VA Filtration, Nuriootpa, Australia)	Qf = ns Qs = ns T = 55 P = 30 Time = 90	17.0	14.5	Alcohols Acids Esters	8.2 15.9 17.4		[169]
	Cabernet Sauvignon red wine B	Spiral wound 4040 and hollow fiber perstractive membrane (VA Filtration, Nuriootpa, Australia)	Qf = ns Qs = ns T = 55 P = 30 Time = 90	15.5	13.3	Alcohols Acids	3.8 12.0		[169]
	Cabernet Sauvignon red wine C	Spiral wound 4040 and hollow fiber perstractive membrane (VA Filtration, Nuriootpa, Australia)	Qf = ns Qs = ns T = 55 P = 30 Time = 90	14.9	13.3	Alcohols	16.4		[169]
	Cabernet Sauvignon red wine D	Spiral wound 4040 and hollow fiber perstractive membrane (VA Filtration, Nuriootpa, Australia)	Qf = ns Qs = ns T = 55 P = 30 Time = 90	14.5	13.2	Alcohols Acids Esters	7.1 4.7 76.5		[169]

Co = original alcohol content; Cf = final alcohol content; T = temperature; P = pressure; VP = vacuum pressure; PP = polypropylene; ns = not specified; Verdicchio white wine 1 = sample 1 of 3; Cabernet Sauvignon red wine A = sample 1 of 5; OD = osmotic distillation; EP = evaporative perstraction; SCC = spinning cone column; NF = nanofiltration; RO = reverse osmosis; PV = pervaporation; PDMS = polydimethylsiloxane; unc = unchanged; nf = not found; *ethanol content removal between 2% and 4% v/v; **no values of the individual volatile aroma compound losses were provided; SPE = solid phase extraction; GC = gas chromatography; MS = mass spectrometry; LE = liquid extraction; FID = flame ionization detector; SBSE = stir bar sorptive extraction; HS = headspace; SPME = solid phase micro extraction; – means not applicable. Units: Concentration = (%v/v); Vacuum pressure/Pressure = bar; Rejection = %; T = °C; Flowrate = L/min; Time = min.

**Table 4 foods-10-02498-t004:** Summary of the main results of some studies on the sensory changes caused by the removal of ethanol from wine by various dealcoholization processes.

Dealcoholization Process	Wine Type	Membrane	Operating Mode/Conditions	Alcohol Reduction		Findings on Sensory Characteristics	Reference
				Co (% v/v)	Cf (% v/v)		
NF	Red Wine	Polyamide, NF97, NF99 HF Alfa Laval	T = 30 P = 16	12.0	9.1	Increase in astringency and unbalanced aroma and taste due to alcohol reduction	[33]
RO	Syrah red wine	ns	T = ns P = ns	12.7	11.1–9.6	Decrease in wine length in the mouth and increase in red fruits and then woody and blackcurrant perceptions (using TDS and attributed to alcohol reduction). Decrease in heat and sweetness intensity (attributed to alcohol reduction) and red fruit intensity (attributed to RO)	[189]
	Merlot red wine	ns	T = ns P = ns	13.4	11.8–10.2	Decrease om wine length in the mouth and increase in astringent and then of fruity perceptions (using TDS and attributed to alcohol reduction). Decrease in heat and texture intensity (attributed to alcohol reduction) and increase in acid intensity (attributed to RO)	[189]
	Syrah red wine	ns	T = ns P = ns	13.4	11.4–7.9	Decrease in persistence, complexity, number of aromas and increase in balance, harmony, and familiarity. Decrease in familiarity and harmony after 4% v/v reduction	[244]
OD/EP	white wine	PTFE hollow fiber (Teflon, Verona, Italy)	Qf = 0.2 Qs = 0.2 T = 20 Time = ns	ns	*	Floral, fruity, and vegetable notes, as well as acidity, saltiness, and bitterness, were not significantly influenced. Decrease in wine body, persistence, and honey note.	[196]
	Falanghina white wine	Liqui-Cel 0.5x1, PP hollow fiber	Qf = 0.07 Qs = 0.14 T = 10 Time = 240	12.5	9.8–0.3	Decrease in odor, sweetness, and body, resulting in unbalanced taste and overall unacceptable, with an unpleasant aftertaste	[36]
	Aglianico red wine	Liqui-Cel Extra-flow, PP hollow fiber	Qf = 0.583 Qs = 0.183 T = 20 Time = 283	13.8	11.6–8.8	Decrease in cherry, red fruits, and sweet notes. Increase in flowers notes only within 2% v/v reduction. Increase in grass and cooked notes and increase in astringency within 5% v/v reduction. Increase in bitterness and acid sensations within 3% v/v reduction	[32]
	Aglianico red wine	Liqui-Cel Extra-flow	Qf = ns Qs = ns T = ns Time = 180	12.8	4.9–0.4	Decrease in sweet and solvent aroma series (due to alcohol reduction) which characterize the wine	[197]
	Aglianico red wine	Liqui-Cel Extra-flow, PP hollow fiber	Qf = 0.583 Qs = 0.183 T = 20 Time = 283	15.5	13.5–10.8	Decrease in cherry, red fruits, flowers, and grass notes. Increase in acid and astringent sensations	[32]
	Montepulciano d’Abruzzo red wine	Liqui-Cel 0.5×1, PP hollow fiber	Recycling mode Qf = 1.5 Qs = 0.5 T = 10 Time = 240	13.2	8.3–2.7	Increase in acidity, a decrease in red fruits and spices notes, astringency, bitterness, and sweetness, resulting in lower acceptability	[35]
PV	Cabernet Sauvignon red wine	PDMS JS-WSM-8040 (JiuSi High-Tech, Nanjing, China)	Batch operation T = 45 VP = 0.05	12.5	0.5	High retention of fruit aroma, producing wine with better smell and taste	[210]
SCC	Chardonnay white wine	–	ns	14.9	14.6–12.9	Decrease in overall aroma intensity and hot mouthfeel sensation	[245]
RO-OD/EP	Shiraz red wine	Memstar AA MEM-074 and Liqui-Cel 2.5 × 8 Extra-flow PP hollow fiber	Qf = ns Qs = ns T = ns P = ns Time = ns	16.3	13.3–10.4	Increase in dark fruit, raisin/prune, alcohol, and astringency in all dealcoholized wines with no significant effects. Increase in black pepper note and overall aroma intensity, and decrease in herbaceous note within 6% v/v reduction off alcohol	[217]
	Cabernet Sauvignon red wine A	Spiral wound 4040 and hollow fiber perstractive membrane (VA Filtration, Nuriootpa, Australia)	Qf = ns Qs = ns T = 55 P = 30 Time = 90	17.0	14.5	Increase in dark fruit aroma and decrease of green aroma, dried fruit, and chocolate flavors with no significant difference in the overall intensity. A small decrease in acidity. Small but significant decreases in sweetness and saltiness. Increase in the sensation of astringency	[169]
	Cabernet Sauvignon red wine B	Spiral wound 4040 and hollow fiber perstractive membrane (VA Filtration, Nuriootpa, Australia)	Qf = ns Qs = ns T = 55 P = 30 Time = 90	15.5	13.3	Decreases in hotness, bitterness, and body (attributed to lower ethanol level). Decrease in confection and ‘chocolate’ aromas. Significant decrease in the overall flavor intensity (largely due to the decreased intensity of dark fruit, sweet spice, and chocolate flavors) with no significant effect on the overall intensity	[169]
	Cabernet Sauvignon red wine C	Spiral wound 4040 and hollow fiber perstractive membrane (VA Filtration, Nuriootpa, Australia)	Qf = ns Qs = ns T = 55 P = 30 Time = 90	14.9	13.3	Decrease in hotness (attributed to lower ethanol level). Decrease in confection, dried fruit, and chocolate aromas with no significant difference in the overall intensity. Decrease in the sensation of astringency	[169]
	Cabernet Sauvignon red wine D	Spiral wound 4040 and hollow fiber perstractive membrane (VA Filtration, Nuriootpa, Australia)	Qf = ns Qs = ns T = 55 P = 30 Time = 90	14.5	13.2	Decrease in hotness (attributed to lower ethanol level). Increase in red fruit aroma with no significant difference in the overall intensity	[169]
	Cabernet Sauvignon red wine E	Spiral wound 4040 and hollow fiber perstractive membrane (VA Filtration, Nuriootpa, Australia)	Qf = ns Qs = ns T = 55 P = 30 Time = 90	16.0	14.2	Decrease in hotness (attributed to lower ethanol level). Decrease in overall flavor intensity with no significant difference in the overall intensity. Small but significant decreases in sweetness and saltiness	[169]

Co = original alcohol content; Cf = final alcohol content; T = temperature; P = pressure; VP = vacuum pressure; PP = polypropylene; ns = not specified; Cabernet Sauvignon red wine A = sample 1 of 5; OD = osmotic distillation; EP = evaporative perstraction; SCC = spinning cone column; NF = nanofiltration; RO = reverse osmosis; PV = pervaporation; PDMS = polydimethylsiloxane; unc = unchanged; *ethanol content removal between 2% and 4% v/v. Units: Concentration = (%v/v); Vacuum pressure/Pressure = bar; Rejection = %; T = °C; Flowrate = L/min; Time = min.

## Data Availability

Not applicable.

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
