# Peer review of "Techniques for Dealcoholization of Wines: Their Impact on Wine Phenolic Composition, Volatile Composition, and Sensory Characteristics"

_foods, 2021, doi:10.3390/foods10102498_

Round 1

Reviewer 1 Report

The manuscript "Techniques for dealcoholization of wines: their impact on wine phenolic composition, volatile composition, and sensory characteristics during ethanol reduction” by Faisal Eudes Sam and colleagues show us an interesting approach and review about the impact of alcohol reduction in wines on the quality of wines. The paper is well structured and presents a lot of information on a very relevant topic in current oenology.

Thus, after a careful reading, in my opinion there are only minor points that could be improved.

So, here are some suggestions for analysis and review by the authors:

Line 33-34: Revise the sentence. Suggestion: “Wine ...... from different fruits, namely berries, apples, ....”.

Line 37-38: This is not totally true. Depends de legislation of each country. There are wines with 15% alcohol by volume that are not fortified wines.

Lines 44-46: Wines are also classified according to sugar content.

Line 47-51: It is important to note that the majority of these studies were made in “in-vitro” systems.

Table 1: The title used it is not clear and don’t represent the information show in the table. I suggest for example: Different techniques to reduce wine alcohol content in the several stages of wine production.

Line 143-144: Vitis vinifera in italic style.

Line 150-156: Specify whether it is applicable to red or white grape varieties.

Line 166-169: It will be useful to refer examples of important wine countries where this practice is legal and no legal.

Line 276-283: Wines with difficult microbiological stabilization and storage.

Line 310: “... of different NF process....”.

Figure 2: “Scheme of different ....”.

Line 494: (+)-catechin and (-)-epicatechin.

Table 2 - Title: “compounds using different dealcoholization process”.

Line 512-513: change for (+)-catechin. trans and cis in italic style.

Line 512-513: “anthocyanins (malvidin-3-glucoside)”. Anthocyanins are not only malvidin. Explain better and rewrite.

Table 3 - Title: “.... compounds using different dealcoholization process”.

Table 4 - Title: “ ..... of wine ethanol using different process”.

Table 5 - first column: Change “Name” for “Commercial name”.

Author Response

Point 1:  Line 33-34: Revise the sentence. Suggestion: “Wine ...... from different fruits, namely berries, apples, ....”.

Response 1: The sentence has been revised as “Wine is a fermented beverage popularly produced from different fruits, including berries, apples, and mangoes”.

Point 2:  Line 37-38: This is not totally true. Depends de legislation of each country. There are wines with 15% alcohol by volume that are not fortified wines.

Response 2: The statement has been re-written appropriately.

Point 3: Lines 44-46: Wines are also classified according to sugar content.

Response 3: This has been added to the classification of wines.

Point 4: Line 47-51: It is important to note that the majority of these studies were made in “in-vitro” systems

Response 4: This has been made clear in the sentence (Line 47-51).

Point 5: Table 1: The title used it is not clear and don’t represent the information show in the table. I suggest for example: Different techniques to reduce wine alcohol content in the several stages of wine production.

Response 5: The title of Table 1 has been changed to “Different techniques to reduce wine alcohol content in the several stages of wine production.”

Point 6: Line 143-144: Vitis vinifera in italic style.

Response 6: The name “Vitis vinifera” has been changed to italic style “Vitis vinifera”.

Point 7: Line 150-156: Specify whether it is applicable to red or white grape varieties.

Response 7: The application of early grape harvest in reducing wine alcohol content in wines produced from red or white grape varieties has been specified.

Point 8: Line 166-169: It will be useful to refer examples of important wine countries where this practice is legal and no legal.

Response 8: Examples of important wine countries where the addition of water to grape must is legal or illegal has been mentioned.

Point 9: Line 276-283: Wines with difficult microbiological stabilization and storage.

Response 9: The sentence “Wines with a high residual sugar content may require additional finishing processes such as sulfur dioxide (SO2) preservation or thermal pasteurization to extend shelf life” has been changed to “Wines with difficult microbiological stabilization and storage may require additional finishing processes such as sulfur dioxide (SO2) preservation or thermal pasteurization to extend shelf life”.

Point 10:  Line 310: “... of different NF process....”.

Response 10: The sentence “Figure 2 presents the scheme of NF process for wine dealcoholization” has been re-written as “Figure 2 presents the scheme of different NF process for wine dealcoholization”.

Point 11:  Figure 2: “Scheme of different ....”.

Response 11: The word “different” has been added to the caption of Figure 2.

Point 12: Line 494: (+)-catechin and (-)-epicatechin.

Response 12: “catechin” and “epicatechin” have been re-written as “(+)-catechin” and “(-)-epicatechin”.

Point 13: Table 2 - Title: “compounds using different dealcoholization process”.

Response 13: The tile of Table 2 has been changed from “Some reported changes in wine phenolic compounds following dealcoholization” to “Some reported changes in wine phenolic compounds using different dealcoholization processes”.

Point 14: Line 512-513: change for (+)-catechin. trans and cis in italic style.

Response 14: “(þ)-catechin” and “trans- and cis- resveratrol” have been changed to “(+)-catechin” and “trans- and cis- resveratrol”.

Point 15: Line 512-513: “anthocyanins (malvidin-3-glucoside)”. Anthocyanins are not only malvidin. Explain better and rewrite.

Response 15: The sentence has been re-written for better understanding. “Malvidin-3-glucoside” was the main anthocyanin reported in that study which was minimally affected with the tendency to increase after dealcoholization by SCC.

Point 16: Table 3 - Title: “.... compounds using different dealcoholization process”.

Response 16: The title of Table 3 has been changed to “Some reported changes in wine volatile aroma compounds using different dealcoholization processes”.

Point 17: Table 4 - Title: “ ..... of wine ethanol using different process”.

Response 17: The phrase “using different processes” has been added to the title of Table 4.

Point 18: Table 5 - first column: Change “Name” for “Commercial name”.

Response 18: The table has been removed based on the second reviewer’s suggestion.

Reviewer 2 Report

Abstract:

The sentence „The focus of most winemakers and researchers over the past years has been geared to-12 wards the partial or total dealcoholization of wines…“ should be rephrased since it is not correct that it was in the focus of most winemakers and researchers but only some of them (I would say a minority of them).

Introduction:

L.36 -39. „…and they can also be classified as table (red, rose, or white), sparkling, 36 or fortified based on their alcohol level [1]. Table wines are unfortified wines that contain 37 less than 14% alcohol by volume and are typically served with food [2]“

This categorization is not common in all parts of the world, for example, since recently European wine regulation does not have a category of „table wine“, and no such strict limits related to the alcohol content.  Please modify this sentence.

  1. 43 – 46. Sentence: „Based on alcoholic content, wines can 43 further be classified as alcohol-free (< 0.5% v/v), low-alcohol (0.5 to 1.2% v/v), reduced-44 alcohol (1.2-5.5 or 6.5% v/v), lower-alcohol (5.5–10.5% v/v), and alcoholic wines (> 10.5% 45 v/v) [5,6].“ is again only partially true, and does not cover the regulation related to for example of European and some other regulations related to alcohol content. This is important for an introduction to this subject, and to explain the motivation of industry for dealcoholization. Please provide more details to give insight into the current situation in the most important wine-producing countries.

  1. 57. – 58. Please rephrase this sentence since it contradicts the previously presented results from different studies. „However, wine is commonly regarded as harmful to human health, regardless of consumption levels because it 58 contains alcohol [16].“

  1. 63. „…higher import duties and taxes [20].“ – this is only true for some countries. Please rephrase.
  2. 65. to 67. – Please rephrase this sentence in a more „scientific“ style.

Techniques for Wine Alcohol Reduction

  1. 140. …Pinot Noir (hybrid sp.) wine grapes.“ ? Correct is Corot Noir.
  2. 147. „…blackberry…“ is not a common expression – use „red grape“ or some other commonly used name.

Table 2.

For RO  „No significant differences were observed in total anthocyanins and phenolic com-pounds for both original and dealcoholized wines. Colour intensity increase by around 20% in both stages of dealcoholized wines due to greater amounts of anthocyanins and the tonality diminishes around 15%“ It is a somewhat contradictory statement, please rephrase.

Related to table 2. and the following paragraph (L. 510 – 525) provide some explanation for the results of different studies often reporting an increase of several classes of phenolic compounds (often anthocyanins) after different alcoholization methods.

Table 3.

Please reorganize the table to avoid the redundancy of information presented (group similar wines, results, methods). I suggest that focus is given to wines and to exclude ciders and other beverages.

Table 4.

Same comment as for table 3. please provide the common effect of the certain process on sensory characteristics and try to reorganize the table to avoid redundancy.

Table 5.

I think this table should be excluded. It is not clear which criteria were used for the presentation of selected producers…and in any way, it can be shortly described in one sentence without the names of the producers.

Additional comment – for this type of review it would be useful one paragraph dedicated to the current state and limitations related to labeling of low alcohol and alcohol-free wines.

Author Response

Response to Reviewer 2 Comments

Point 1:  Abstract: The sentence „The focus of most winemakers and researchers over the past years has been geared to-12 wards the partial or total dealcoholization of wines…“ should be rephrased since it is not correct that it was in the focus of most winemakers and researchers but only some of them (I would say a minority of them).

Response 1: The sentence has been rephrased as “The attention of some winemakers and researchers over the past years has been drawn towards the partial or total dealcoholization of wines and alcoholic beverages due to trends in wine styles, and the effect of climate change on wine alcohol content.”

Point 2:  Introduction: L.36 -39. „…and they can also be classified as table (red, rose, or white), sparkling, 36 or fortified based on their alcohol level [1]. Table wines are unfortified wines that contain 37 less than 14% alcohol by volume and are typically served with food [2]“ This categorization is not common in all parts of the world, for example, since recently European wine regulation does not have a category of „table wine“, and no such strict limits related to the alcohol content.  Please modify this sentence.

Response 2: The sentence has been modified.

Point 3: 43 – 46. Sentence: „Based on alcoholic content, wines can 43 further be classified as alcohol-free (< 0.5% v/v), low-alcohol (0.5 to 1.2% v/v), reduced-44 alcohol (1.2-5.5 or 6.5% v/v), lower-alcohol (5.5–10.5% v/v), and alcoholic wines (> 10.5% 45 v/v) [5,6].“ is again only partially true, and does not cover the regulation related to for example of European and some other regulations related to alcohol content. This is important for an introduction to this subject, and to explain the motivation of industry for dealcoholization. Please provide more details to give insight into the current situation in the most important wine-producing countries.

Response 3: The classification of wines based on alcoholic content has been re-written and reasons for the motivation of dealcoholization industry are given in detail.

Point 4: 57. – 58. Please rephrase this sentence since it contradicts the previously presented results from different studies. „However, wine is commonly regarded as harmful to human health, regardless of consumption levels because it 58 contains alcohol [16].“

Response 4: The sentence has been rephrased for better understanding without contradictions to previously presented results from different studies.

Point 5: 63. „…higher import duties and taxes [20].“ – this is only true for some countries. Please rephrase.

Response 5: The sentence has been rephrased.

Point 6: 65. to 67. – Please rephrase this sentence in a more „scientific“ style.

Response 6: The sentence has been rephrased in a more scientific style.

Point 7: Techniques for Wine Alcohol Reduction

  1. …Pinot Noir (hybrid sp.) wine grapes.“ ? Correct is Corot Noir.

Response 7: The name “Pinot Noir (hybrid sp.)” has been changed to “Corot Noir”.

Point 8: 147. „…blackberry…“ is not a common expression – use „red grape“ or some other commonly used name.

Response 8: the word “blackberry” has been changed to “red grape”.

Point 9: Table 2.

For RO  „No significant differences were observed in total anthocyanins and phenolic compounds for both original and dealcoholized wines. Colour intensity increase by around 20% in both stages of dealcoholized wines due to greater amounts of anthocyanins and the tonality diminishes around 15%“ It is a somewhat contradictory statement, please rephrase.

Response 9: The statement has been rephrased as “No significant differences were observed in total anthocyanins and phenolic compounds for both original and dealcoholized wines. Colour intensity increase by around 20% in dealcoholized wines, while the tonality diminished by 15%.

Point 10:  Related to table 2. and the following paragraph (L. 510 – 525) provide some explanation for the results of different studies often reporting an increase of several classes of phenolic compounds (often anthocyanins) after different alcoholization methods.

Response 10: In relation to Table 2 and the following paragraph (L. 510 – 525), explanations for the results of different studies that reported an increase of several classes of phenolic compounds (often anthocyanins) after dealcoholization has been added to the paragraph.

Point 11:  Table 3. Please reorganize the table to avoid the redundancy of information presented (group similar wines, results, methods). I suggest that focus is given to wines and to exclude ciders and other beverages.

Response 11: The wine type, dealcoholization processes (methods), and results are all headings and in organizing the table, only one of these headings can be used as the basis for the presentation of information on the table. Since the focus of this review is on the different dealcoholization methods, we have reorganized the table based on the methods (from NF to RO–OD/EP) and the type of wine (under each method, with model wine first, followed by white, red and rose wines where applicable). Also, ciders have been excluded from the table.

Point 12: Table 4. Same comment as for table 3. please provide the common effect of the certain process on sensory characteristics and try to reorganize the table to avoid redundancy.

Response 12: The table has been reorganized based on the methods (from NF to RO–OD/EP) and the type of wine (under each method, with model wine first, followed by white, red and rose wines where applicable). From the results, a common effect (decrease in hotness) was observed in most of the findings after dealcoholization with different methods. However, we think it is inappropriate to indicate this as a common effect for a particular method or methods because the alcohol reduction levels are different in the studies conducted and, in some studies too, hotness was not part of the sensory attributes.

Point 13: Table 5. I think this table should be excluded. It is not clear which criteria were used for the presentation of selected producers…and in any way, it can be shortly described in one sentence without the names of the producers.

Response 13: The table has been removed from the manuscript.

Point 14: Additional comment – for this type of review it would be useful one paragraph dedicated to the current state and limitations related to labeling of low alcohol and alcohol-free wines.

Response 13: A sub-heading relating to labelling of low- and non-alcoholic wine beverages (including the current state and limitations) was part of the initial writeup of this review. However, there was scarcity of relevant information on it so we had to leave it out. We hope to find relevant information on it in our next review paper in the coming years.

Reviewer 3 Report

Nicely written review, no need to change except maybe a title...Techniques for dealcoholization of wines: their impact on wine phenolic composition, volatile composition, and sensory characteristics    "during ethanol reduction" - this part is redundant

Author Response

Point 1: Nicely written review, no need to change except maybe a title...Techniques for dealcoholization of wines: their impact on wine phenolic composition, volatile composition, and sensory characteristics    "during ethanol reduction" - this part is redundant

Response 1: The title has been changed to “Techniques for dealcoholization of wines: their impact on wine phenolic composition, volatile composition, and sensory characteristics”.
